# Impact of Nutritional Status on Clinical Outcomes of Patients Undergoing PRGF Treatment for Knee Osteoarthritis—A Prospective Observational Study

**DOI:** 10.3390/nu17193134

**Published:** 2025-09-30

**Authors:** Paola De Luca, Giulio Grieco, Simona Landoni, Eugenio Caradonna, Valerio Pascale, Enrico Ragni, Laura de Girolamo

**Affiliations:** 1IRCCS Ospedale Galeazzi Sant’Ambrogio, Orthopaedic Biotechnology Laboratory, 20157 Milan, Italy; deluca.paola@grupposandonato.it (P.D.L.); giulio.grieco@grupposandonato.it (G.G.); simona.landoni@grupposandonato.it (S.L.); laura.degirolamo@grupposandonato.it (L.d.G.); 2Centro Diagnostico Italiano, Department of Clinical Laboratory, 20100 Milan, Italy; eugenio.caradonna@cdi.it; 3IRCCS Ospedale Galeazzi Sant’Ambrogio, Ortopedia Clinicizzata, 20157 Milan, Italy; valerio.pascale@unimi.it; 4Università degli Studi di Milano, Scienze Biomediche per la Salute, Medicina e Chirurgia, 20122 Milan, Italy

**Keywords:** osteoarthritis, platelet-rich growth factors, nutrition, BMI

## Abstract

**Background:** Osteoarthritis (OA) is a major global health issue, increasing with aging and obesity. Current therapies mainly address symptoms without modifying disease progression. Platelet-rich growth factor (PRGF) therapy has potential regenerative effects through high cytokines and growth factors, but the outcomes of these therapies remain heterogeneous. This study explores the relationship between patient nutritional status, PRGF characteristics, and clinical outcomes in knee OA treatment. **Methods:** Baseline anthropometric, metabolic, and nutritional assessments of 41 patients with knee OA who underwent PRGF treatment were conducted. Blood samples were analyzed for metabolic and inflammatory markers. PRGF composition was assessed by protein content and extracellular vesicle (EV) markers. KOOS and VAS pain scores were collected at 2, 6, and 12 months. Responders improved KOOS by ≥10 points. An elastic-net regularized logistic model allowed the identification of the predictors of treatment response. **Results:** KOOS and VAS scores improved significantly at all follow-ups. At 2 months, the PRGF of responder patients showed higher PRGF G-CSF levels; at 12 months, increased CD49e and HLA-ABC expression. Higher BMI correlated with increased IL-6, IL-1ra, and resistin in PRGF samples. Hypercholesterolemic patients displayed altered EV profiles, with elevated levels of CD8 but reduced CD49e, HLA-ABC, CD42a, and CD31. Multivariate analysis identified BMI, biceps fold, fat percentage, red blood cell, platelet, and neutrophil counts as predictors of early response. **Conclusions:** Metabolic and immunological factors influence PRGF composition and clinical efficacy in knee OA. Baseline body composition and hematological parameters as key predictors of response, highlighting the potential of personalized PRGF therapy.

## 1. Introduction

Osteoarthritis (OA) is a prevalent global health issue, further aggravated by the aging population and increasing obesity rates [1], affected an estimated 7.6% of the global population [2]. It is characterized by progressive joint impairment with articular cartilage degradation, synovial inflammation, ligament degeneration, and osteophyte formation. The result is chronic pain and functional restrictions with significant decline in quality of life [2]. Increasing evidence underscores mechanical loading as a significant risk factor in OA, which activates mechano-sensitive signaling pathways [3]. The knee joint is most commonly affected, with a global prevalence of 22.9% in individuals over 40 years of age [4].

Current therapeutic strategies for OA predominantly focus on pain management, with recommendations for oral non-steroidal anti-inflammatory drugs (NSAIDs), analgesics, symptomatic slow-acting drugs for OA (SYSADOA), and non-pharmacological approaches, such as physical therapy. While these treatments offer short-term symptomatic relief, no current therapy has been shown to halt the progressive degeneration of the joint [5]. Indeed, due to limited understanding of the pathogenesis and the multifactorial etiology of this heterogeneous disease, there are currently no disease-modifying therapies available [6].

One potential therapeutic approach for osteoarthritis involves the use of orthobiologics, including autologous products such as platelet-rich plasma (PRP), bone marrow aspirate concentrate (BMAC), adipose tissue stromal vascular fraction (SVF), and micro-fragmented adipose tissue (MFAT). However, existing studies have demonstrated significant heterogeneity in the reported outcomes [7].

Blood-derived products are the most extensively studied category of orthobiologics due to their easy availability and high concentrations of growth factors and cytokines, which have the potential to promote tissue homeostasis restoration [8,9]. Despite this, a substantial proportion of patients remain unresponsive to treatment. While variations in preparation protocols, resulting in differences in platelets, growth factors, cytokines, leukocyte concentrations, and added components have been proposed as contributing factors [10,11], emerging evidence suggests that individual baseline patient characteristics, particularly those related to blood composition and health, may play an even more significant role.

Factors such as baseline platelet concentration and function have been shown to critically influence treatment efficacy, with higher platelet counts in the administered product correlating with better clinical outcomes [12]. Furthermore, genetic polymorphisms affecting blood components and coagulation profiles can further modulate how patients respond to therapy [13]. These intrinsic blood-related factors, also influenced by diet and other lifestyle elements, alongside other biological and clinical variables such as inflammatory status and possible disease severity, highlight the importance of considering patient-specific variables when evaluating therapeutic efficacy and suggest that a more personalized approach may be necessary to optimize results.

To better understand the factors influencing the heterogeneous patient’s response and considering the direct impact of metabolic and nutritional status on the quality of blood, this study aims to assess the relationship between patients’ nutritional status, characteristics of blood-derived products, and clinical outcomes in knee osteoarthritis patients undergoing PRGF treatment.

## 2. Materials and Methods

### 2.1. Study Design, Patient Enrolment, and Data Collection

In this prospective, monocentric, observational, non-controlled study (case series), 41 consecutive patients (mean age 50.8 ± 10.2 years, twenty-five males and sixteen females) affected by moderate to severe knee osteoarthritis (Kellgren–Lawrence II–III) and undergoing injective treatment with PRP-Endoret^®^ (PRGF^®^) (BTI, Vitoria, Álava, Spain) (two injections two weeks apart) were recruited at REGAIN, the center for regenerative medicine at the IRCCS Istituto Ortopedico Galeazzi of Milan, upon the approval of the San Raffaele Hospital Ethics Committee (CE:125/INT/2021, approved on 14 July 2021). Following the acquisition of informed consent, patients underwent an assessment of their nutritional status by the same nutritionist. Subsequently, blood samples were collected, and PRGF injections were administered by an orthopedic specialist.

Patient-reported outcome measures (PROMs) were collected before the treatment and at 2, 6, and 12-month follow-ups through an online survey system tailored for the study and based on an established survey platform (QualtricsXM, Qualtrics, Provo, UT, USA) already in use at REGAIN. Specifically, patients were administered the validated Italian version of the Knee Injury and Osteoarthritis Outcome Score (KOOS) [14], which constitutes the primary outcome, and the Visual Analog Scale for pain (VAS). The KOOS possible score ranged from 0 to 100 (worst–best), and the VAS score ranged from 0 to 10 (best–worst).

Due to substantial loss to follow-up, analyses at each time point (2, 6, 12 months) were conducted on the available cases only. This approach was chosen due to the extent of missing and limited information, which would not allow for reliable imputation.

### 2.2. Anthropometric, Metabolic, and Nutritional Evaluations

Body mass index (BMI), body composition (% free fatty mass, FFM, and % fatty mass, FM), and anthropometric measurements were assessed by the same nutritionist the day of the enrollment, just prior to the first injection (T0). All equipment was calibrated as recommended by the manufacturers. Body height was measured to the nearest 0.1 cm with a stadiometer (SECA) with participants wearing light clothing. Body weight was measured to the nearest 0.1 kg with a personal weighing scale (SECA).

Patients with a BMI between 18.5 and 24.9 kg/m^2^ were considered normal weight, while patients with a BMI over 24.9 kg/m^2^ were considered overweight.

The percentage of fatty mass and free fatty mass were calculated through plicometry and using the Durnin–Womersley formula [15]. The four skinfolds (biceps, triceps, subscapular, and supracrestal) were measured using a caliper (Holtain Ltd., Crymych, UK) with participants standing in the anatomical position. Each skinfold was measured three times, and the average of these measurements was used for all analyses.

At the time of the first injection, patients’ blood samples were collected and analyzed in the laboratory.

Additionally, patients completed an Italian version of dietary history questionnaires (nutritional anamnesis, Mediterranean diet index) consisting of questions identifying daily caloric intake, eating habits, presence of diseases, and the use of medications or supplements.

The Mediterranean diet index consisted of nine food categories: fruit, vegetables, cereal grains, legumes, fish and fish products, meat and meat products, dairy products, alcohol intake, and olive oil. For food groups characteristic of the Mediterranean diet (fruit, vegetables, cereals, legumes, and fish), the highest consumption category was assigned a score of 2, the middle category a score of 1, and the lowest category a score of 0. For food groups not typical of the Mediterranean diet (meat, meat products, and dairy), the lowest consumption category received a score of 2, the middle category a score of 1, and the highest category a score of 0. For alcohol, categories based on alcohol units (1 unit = 12 g of alcohol) were used, with 2 points assigned for 1–2 units per day, 1 point for 1 unit per day, and 0 points for higher consumption. For olive oil, 2 points were given for regular use, 1 point for frequent use, and 0 points for occasional use. The final score, representing adherence to the Mediterranean diet, was the sum of these values, ranging from 0 (low adherence) to 18 (high adherence) [16].

### 2.3. PRGF Collection

Platelet-rich growth factor (PRGF) was obtained following the manufacturer’s instructions for the PRGF–Endoret system [17]. In detail, 18 mL of peripheral blood was collected by a research nurse using 21-gauge needles and placed in two 9 mL sterile extraction tubes containing 3.8% trisodium citrate. After centrifugation at 2100 rpm for 8 minutes at room temperature, the blood was separated into different phases: plasma (yellow portion), buffy coat (rich in leukocytes), erythrocytes (red portion). The upper portion of the plasma was indicated as platelet-poor plasma (F1), while the lower portion was referred to as platelet-rich plasma (F2). To activate platelets, 0.8 mL of 10% CaCl_2_ was added to the F2 fraction (final CaCl_2_ concentration of 22.5 μM), and the resulting preparation (4 mL) was injected into the patient’s knee.

A small volume of F2 was kept for hematological analysis, whereas, for research purposes only, the portion of the F1 fraction closest to F2 was activated with CaCl_2_ 225 mM (final concentration 22.5 μM) and centrifuged at 2800× *g* for 15 min at room temperature. The resultant supernatant was recovered, yielding PRGF that was subsequently characterized.

### 2.4. Isolation of EVs from PRGF Samples

Isolation of EV (size 30–200 nm) from human PRGF samples was performed with the exosome purification kit, Exo-SpinTM blood (Cell Guidance Systems, Cambridge, UK), in accordance with the producer’s protocol. Briefly, 250 µL of PRGF were centrifuged at 300× *g* for 10 min at room temperature (RT), to remove unwanted cells and remaining platelets. A subsequent supernatant centrifugation at 16,000× *g* for 30 min removed small debris. The supernatant was added with ExospinTM Buffer in a 2:1 ratio (two volumes of supernatant + one volume of buffer) and incubated for 5 min at 4 °C, then centrifugation at 16,000× *g* for 30 min at RT was carried out. The obtained pellet containing EVs was suspended in 100 µL of phosphate-buffered saline (PBS). After column equilibration, the suspension with EVs was added and centrifuged at 50× *g* for 60 s at RT. The column was eluted with 200 µL of PBS to obtain the purified EVs.

After dilution of purified EVs (1:100) with PBS, they were visualized by the NanoSight NS-300 system (NanoSight Ltd., Amesbury, UK) (5 recordings of 60 s). With NTA software, v3.4 concentration and high-resolution particle size distribution profiles were measured.

### 2.5. Nanoparticles Analysis with the Multiplex Bead-Based Platform

EVs of each sample were labeled with MACSPlex EV Kit IO according to the manufacturer’s instructions (Miltenyi Biotec srl, Bologna, Italy). The MACSPlex EV Kit IO consists of a mix of fluorescently labeled bead populations, each coated with a specific antibody that binds to corresponding surface epitopes. These 37 bead populations plus 2 controls (CD3, CD4, CD19, CD8, HLA-DRDPDQ, CD56, CD105, CD2, CD1c, CD25, CD49e, ROR1, CD209, CD9, SSEA-4, HLA-ABC, CD63, CD40, CD62P, CD11c, CD81, MCSP, CD146, CD41b, CD42a, CD24, CD86, CD44, CD326, CD133/1, CD29, CD69, CD142, CD45, CD31, CD20, CD14, REA control, mIgG1 control) can be differentiated based on varying fluorescence intensities using flow cytometry.

Briefly, 120 μL samples containing either buffer (blank control) or 1.5 billion/mL of purified EVs were loaded onto 1.5 mL vials with 15 μL of MACSPlex Exosome Capture Beads and incubated on an orbital shaker overnight (14–16 h) at 450 rpm at RT. Therefore, the beads were washed in PBS and suspended in 135 μL of MACSPlex Buffer (MPB). EVs captured by beads were incubated for 1 h at room temperature with 15 μL of MACSPlex Exosome Detection Reagent cocktail, containing antibody conjugates for CD9/CD63/CD81-APC. Following the incubation, the beads were washed and analyzed using flow cytometry.

### 2.6. Flow Cytometry Analysis

After flow cytometer (Cytoflex, Beckman, Brea, CA, USA) calibration with MACSPlex EV Setup Beads, samples (blank or EVs) were acquired for 15 s in Allophycocyanin (APC). For each tested marker, after blank control fluorescence subtraction, the median fluorescence intensity of the signals detected for the MACSPlex EV Capture Beads CD9, CD63, CD81 was calculated. The mean of these median signals was used as the normalization factor for each sample. The relative level of EV marker was determined by calculating the ratio of the signal intensities of hypercholesterolemic samples vs. normocholesterolemic samples. Markers detected in at least 70% of samples were analyzed.

### 2.7. Analysis of the Selected Proteins Released by Luminex^®^ Assays

A custom panel for the G-CSF, HGF, IL-10, IL1b, IL1ra, IL-6, IL-8, VEGF, P-selectin and resistin multiple detection was created (Human Magnetic Luminex Screening Assay, Bio-Techne, Minneapolis, MN, USA, sensitivity: 4.1 pg/mL, 1 pg/mL, 1.6 pg/mL, 0.8 pg/mL, 18 pg/mL, 1.7 pg/mL, 1.8 pg/mL, 0.99 pg/mL, 9 pg/mL, and 3 pg/mL, respectively).

The assay was performed on a MagPix™ Luminex System (Bio-Rad Laboratories, Inc., Hercules, CA, USA) following the manufacturer’s instructions using a volume of 50 µL for each PRGF obtained.

### 2.8. Statistical Analysis

Analyses were conducted using R software version 4.1.3 (R Core Team, Vienna, Austria), RStudio software version 2024.12.1+563 (Posit Software, PBC), and GraphPad Prism version 9.5.0. Categorical variables are presented as absolute and relative frequencies, while numerical variables are expressed as means and standard deviations. The distribution of continuous variables was assessed using the Shapiro–Wilk test. Depending on the outcome of the normality test, parametric tests (Student’s *t*-test, one-way ANOVA) or non-parametric tests (Mann–Whitney U test, Kruskal–Wallis test, Friedman test) were applied to compare differences across groups and follow-up times. To account for multiple testing, *p*-values obtained from individual *t*-tests were adjusted using the Benjamini–Hochberg false discovery rate (FDR) procedure. Correlation analysis was performed using Pearson or Spearman tests, depending on the results of the normality assessment. Logistic regression models were used to identify factors associated with the categorical outcome (responder), defined as the change in KOOS total score (ΔKOOS) from baseline to subsequent time points. A *p*-value of less than 0.05 was considered statistically significant. A post hoc sensitivity power analysis was carried out to address statistical significance after stratification into subgroups of patients.

The association between the categorical outcome and various covariates, including established predictors and newly investigated anthropometric and blood count variables, was assessed [18,19]. Inter-variable multicollinearity quantified using the Variance Inflation Factor (VIF) [20]. The dataset was split using a partition ratio of 0.6 for the training set and 0.4 for the test set to develop the multivariate logistic regression model. A stepwise selection approach was implemented to optimize model predictability by iteratively removing variables based on descending *p*-values. Variables with a *p*-value < 0.20 in the univariate regression analysis were included in the multivariate regression model. The Elastic Net regularization method that incorporates the L1 and L2 coefficient norms used in Lasso and Ridge regressions was employed for refinement of the best predictive logistic regression model and integrated with nested leave-one-out cross-validation (LOOCV) to maximize the use of the dataset for hyperparameter tuning and external validation [21,22]. The optimal tuning parameter α for the Elastic Net, which governs the balance between L1 and L2 regularization, was chosen for each dataset and its corresponding outcome. The outer loop that resulted from nested cross-validation was exploited as an external validation set to assess the final model’s predictive performance. Model selection was guided by the Akaike Information Criterion (AIC), allowing for comparison between models with varying degrees of complexity [23].

## 3. Results

### 3.1. Collected Data Evaluation and Anthropometric/Nutritional Assessments of Enrolled Patients

As shown in the STROBE-style flow diagram (Figure 1), the number of participants decreased from 41 at baseline to 34 at 2 months, 27 at 6 months, and 23 at 12 months, with a substantial loss to follow-up being observed over time. Therefore, analyses at each time point were carried out on the available cases only.

Anthropometric measurements are reported in Table 1. They included age (years), sex, weight (kg), height (m), BMI (kg/m^2^), waist circumference (cm), arm circumference (cm), biceps fold (mm), triceps fold (mm), subcrestal fold (mm), subscapular fold (mm), FFM%, FM%. The blood values obtained from blood sampling immediately prior to treatment and relative to complete blood count, glucose, creatinine, triglycerides, total cholesterol, high-density lipoproteins (HDL), C-reactive protein (CRP), glycated hemoglobin (A1c), aspartate aminotransferase (AST), and alanine aminotransferase (ALT), along with their respective normal reference ranges, are shown in Appendix A. Of the 41 patients, 21 were of normal weight and 20 were overweight. No statistically significant differences were shown for any of these blood parameters between responsive and nonresponsive patients.

On average, the platelet concentration in the PRGF-injected samples was three times more concentrated than in whole blood (3.09 ± 0.38).

A total of 26 patients (63.4%) exhibited high adherence to the Mediterranean diet, whereas 5 patients (12.2%) showed low adherence (Table 1). The nutritional anamnesis and dietary history showed that no patients had food that is known to affect the quality of the PRGF in the 8 h prior to blood collection.

### 3.2. Evaluation of the Clinical Outcomes

Patients were considered responsive to the PRGF treatment if they demonstrated a 10-point improvement in their KOOS, considered as primary outcome measure, from pre- to post-treatment, as reported in the literature [24].

At two months of follow-up, 44.1% of patients responded to treatment, of which 33.3% normal weight and 66.6% overweight, at six months the percentage of responder patients was 40.7% of which 45.45% normal weight and 54.55% overweight, and of 43.5% at last follow-up of which 55.55% normal weight and 44.45% overweight.

Overall, considering all patients at each time point, KOOS improved significantly over time at each follow-up compared to the baseline values, as shown in Figure 2A. Similarly, the VAS scale showed a significant decrease at each time point compared to the baseline for all patients included in the study. Normal-weight patients showed significant improvements only at 12 months of FU respect to the baseline for both KOOS total and VAS scores. Otherwise, overweight patients showed significant improvement only for the VAS scale; both at 2 and 12 months of FU (Figure 2B).

All data that are relative to KOOS total and its subscale scores at each time point for all patients are reported in Appendix A.

### 3.3. PRGF Characteristics in Different Patient Subgroups

PRGF samples were assessed for their biochemical composition and for their content in EVs. A number of EVs/mL ranging from 5.25 × 10^9^ to 1.77 × 10^11^ (mean 6.0 ± 3.4 × 10^10^) was found in the samples. The mean diameter of EVs ranged from 92.6 to 186.4 nm, while the mean of the mode of EVs was 97.9  ±  17.4 nm.

### 3.4. PRGF Characteristics of Normal Weight and Overweight Patients

By analyzing patients’ characteristics according to BMI values, the differences that emerged between normal weight and overweight patients concerned the levels of IL6, IL1ra, and resistin, which were significantly higher in overweight patients (*p*-value < 0.05) (Figure 3). Given the limited subgroup sample sizes (n = 22 for overweight and n = 19 for normal weight subjects), a sensitivity analysis indicated that only large effect sizes could be detected (Cohen’s d coefficient = 0.899 with 80% power at 0.05 level of significance).

### 3.5. PRGF Characteristics of Responsive and Nonresponsive Patients

Based on patients’ responsiveness to treatment, among the growth factors and cytokines investigated in PRGF samples, the only statistically significant difference was found for G-CSF, which was higher in responders in respect to non-responder patients at 2 months’ follow-up (*p*-value < 0.05).

Regarding EV characterization, those derived from patients who were found to be responsive at two months of follow-up, showed an interesting tendency for higher levels of CD86, CD1c, CD4, CD19, CD20, CD44, and MCSP (Appendix A). EVs of responder and non-responder patients identified at 6 months showed no difference in any of the parameters evaluated, whereas when compared at 12 months, EVs of responder patients had statistical significant higher level of CD49e (*p*-value < 0.01) and HLA-ABC (*p*-value < 0.05) (Figure 4), as well as a tendency for lower levels of MCSP, CD19, CD20, CD86, and CD11c (Appendix A) .

### 3.6. PRGF Characteristics of Normocholesterolemic and Hypercholesterolemic Patients

An evaluation of the characteristics of the normocholesterolemic and hypercholesterolemic patient subgroups revealed that patients with a higher level of total cholesterol (>200 mg/dL) showed EVs with significantly higher levels of CD8 (*p*-value < 0.05), and significant lower level of CD49e, HLA-ABC, CD42a, and CD31 (*p*-value < 0.01), as shown in Figure 5. Given the limited subgroup sample sizes (n = 22 for hypercholesterolemic and n = 19 for normocholesterolemic subjects), a sensitivity analysis indicated that only large-effect sizes could be detected (Cohen’s d coefficient = 0.899 with 80% power at 0.05 level of significance).

Samples derived from hypercholesterolemic patients (58% of the sample) showed a higher number of EVs (mean: 7.62 × 10^10^ ± 3.4 × 10^10^ vs. 3.7 × 10^10^ ± 1.7 × 10^10^) with lower size (mean: 128.7 ± 15.1 nm vs. 152.0 ± 18.1 nm) compared to patients with normo-cholesterol levels (*p*-value < 0.0001 for both parameters).

### 3.7. PRGF Characteristics in Male and Female Patients

Among the cohort of recruited patients, 61% were male and 39% were female. The association between sex and the mean concentrations of cytokines or EV markers in PRGF was evaluated. Statistically significant differences were observed using multivariate analysis of variance (MANOVA). Subsequently, post hoc *t*-test analyses highlighted males exhibiting lower levels of IL-1ra (*p* < 0.05) and higher levels of P-selectin (*p* < 0.05) compared to females (Figure 6A). Additionally, levels of CD2 and CD29 EV markers were significantly elevated in males compared to in females (*p* < 0.05) (Figure 6B). Due to the small sample sizes in the subgroups (n = 25 male and n = 16 female subjects), a sensitivity analysis revealed that only large effect sizes could be detected (Cohen’s d coefficient = 0.919 with 80% power at a significance level of 0.05).

### 3.8. Statistical Modeling and Predictive Factors

#### 3.8.1. Multivariate Logistic Regression Model

The multivariate logistic regression model was performed only at 2 months’ follow-up, when the number of patients who responded to the questionnaire used to assess patient-reported outcomes related to knee injuries and osteoarthritis (KOOS) was adequate.

The Spearman rank correlation coefficient was initially used to assess the relationship between both continuous and categorical variables included in the study. This analysis aimed to identify inter-variable correlations and eliminate those with high Spearman correlation coefficients (|r| ≥ 0.6). Afterward, anthropometric and blood level covariates of each patient were incorporated into a multivariate logistic regression model following a stepwise selection of the most predictive variables. Six clinical covariates were retained: BMI (Kg/m^2^), biceps fold (mm), FM%, red blood cell count (10^12^/L), platelet count (10^9^/L), and neutrophils cell count (10^9^/L). The resulting model exhibited optimal predictive performance, with an area under the curve (AUC) of 0.833 (95% CI: 0.461–0.949) and an AIC value of 44.76 (Figure 7).

Parameter estimates and variable coefficients included in the model are detailed in Table 2.

Following this analysis, the parameters identified in the model with our values using a *t*-test were evaluated. The results showed a trend towards higher values for platelet and neutrophil counts in responsive patients than non-responsive patients (*p*-value 0.0641 and 0.09, respectively). A slight increased level of fatty mass percentage was observed in responders respect to non-responder patients (*p*-value 0.09), while no difference was shown between the two categories of patients with regard to red blood cell count and BMI.

#### 3.8.2. Elastic Net Regularization Method

The coefficient estimate for red blood cell count, as well as the AUC value of the simple logistic regression model, were found to be inflated and overestimated. To improve the predictive performance of the logistic regression model, with respect to the ΔKOOS total score clinical outcome at 2 months after treatment, an Elastic Net regularization approach integrated with LOOCV nested cross-validation was employed. The optimal tuning parameter α was determined through the exploitation of the inner loop set of the nested cross-validation, and it resulted in 0.1, positioning the regularization more proximate to the Ridge technique. The outer loop was employed as an external validation set, resulting in a model performance of an AUC value decreased to 0.691 (95% CI: 0.51–0.87) (Figure 8). This adjustment led to the development of a reduction in predictive performance but the implementation of a more reliable model fit for the data subset used for testing compared to the initial logistic regression model.

## 4. Discussion

The main finding of this study was that metabolic factors associated with the nutritional status of patients impact on the quality of PRGF in terms of cytokine and EV content.

Moreover, for the first time, this study evaluated the effect of nutritional status (anthropometric values, blood-metabolic parameters, and dietary history) on the clinical outcomes of patients undergoing PRGF treatment for knee OA.

Blood-derived therapy has emerged as a promising treatment option for OA, offering potential for modulation of inflammatory and regenerative processes. In particular, PRGF is an autologous blood product enriched with growth factors, cytokines, and extracellular vesicles, which are thought to play a role in tissue repair and regeneration [25,26]. The therapeutic potential of PRGF lies in its ability to deliver high concentrations of growth factors such as platelet-derived growth factor (PDGF), vascular endothelial growth factor (VEGF), insulin-like growth factor (IGF-I), and transforming growth factor-beta (TGF-β), all of which promote tissue repair [27]. However, the clinical outcomes of PRGF therapy have shown considerable variability, with some patients demonstrating significant improvement while others a limited response. This heterogeneity is likely due to several factors, including differences in PRGF preparation protocols, platelet yield variations [28], in growth factor concentrations, and patient-specific factors such as metabolic and nutritional status [10,11].

The results of this study indicated an overall significant improvement in both the KOOS and VAS scores over time in patients treated with PRGF, with results demonstrating the highest response at the 2-month follow-up, where the responsiveness rate of all populations according to the primary outcome (KOOS) was 44.1%. While this percentage slightly changes at the 6-month (40.7%) and 12-month (43.5%) follow-ups, the improvement was sustained, highlighting the potential long-term benefits of this therapy.

An intriguing observation emerges when examining the rate of treatment responsiveness across patients with different BMI levels. In fact, while at early follow-ups overweight patients showed a higher rate of responsiveness with respect to normal-weight patients, as time progresses, this trend reverses. Therefore, patients with a normal BMI exhibited a delayed yet progressive enhancement over time. This can be explained both by the mechanical overload on the knee, as well as by the inflammation milieu induced by obesity, overcoming the effects of PRGF quicker than in normal patients.

In fact, when assessing how the nutritional-metabolic status of patients impacted PRGF characteristics, it was found that patients with higher levels of BMI exhibited higher concentrations of inflammatory cytokines such as IL-6, IL-1ra, and Resistin, suggesting that metabolic and nutritional factors may influence the inflammatory blood milieu in OA patients. These findings are in accordance with the literature showing that factors like adipose deposition, insulin resistance, and especially the improper coordination of innate and adaptive immune responses may lead to the initiation and progression of obesity-associated OA [29,30].

Looking at the characteristics of PRGF samples of the patients of this study and focusing on treatment responsiveness, it was interesting to find that responder patients showed higher levels of G-CSF. This could be attributed to the involvement of this growth factor in the repair processes of the cartilage tissue, as previously demonstrated [31].

In recent years, in the field of regenerative medicine, particular attention has been taken to the study of EVs. This is due to their different properties, functions and compositions, and their role in tissue repair. In platelet-based blood derived products, cytokines derived from platelets are mainly present in exosomes. Our study revealed significant differences in the expression of immune cell markers on EVs between responder and non-responder patients.

At 12 months follow-up, a higher level of the chondroprogenitor marker CD49e, also referred to as the fibronectin receptor, was detected [32]. The extracellular matrix degradation, typically observed in OA, results in the release of matrix-derived protein fragments, such as fibronectin, capable of interacting with transmembrane integrin receptors. It was demonstrated that this interaction disrupts cellular signaling pathways, thereby facilitating disease progression [33,34,35,36,37,38]. The binding between the fibronectin receptor CD49e on EVs and fibronectin fragments may help to break the cycle in which these matrix fragments exacerbate cartilage degradation, potentially explaining the enhanced responsiveness observed in patients with higher CD49e levels. In addition, the significantly lower level of CD49e, CD42a, and HLA-ABC in hypercholesterolemic patients may suggest both a reduced ability to counteract cartilage degradation and lower platelet activation [39]. Indeed, HLA class I antigens on platelets play a role in platelet aggregation and subsequent release of growth factors [40]. This also justifies the significantly higher value of human leukocyte-abacavir antigen (HLA-ABC) in samples derived from responder patients at 12-month follow-up.

As expected, the blood total cholesterol level was a differentiating element in the quality of the released EVs [41].

The higher value of typical lymphocyte markers CD8 in hypercholesterolemic patients suggests a higher inflammatory status of these subjects [42]. The platelet endothelial cell adhesion molecule CD31, also referred to as PECAM1, is normally expressed or reduced in cases of hypercholesterolemia [43], while in normocholesterol patients, the higher concentration of CD31 may be associated with a more severe degree of OA, as previously reported [44,45,46]. These findings suggest that an immune-regulatory balance may play a role in treatment success, underlining the need for personalized approaches based on individual immune profiles.

Taken together, these results highlight the need for a comprehensive approach to patient assessment, which considers not only clinical symptoms but also metabolic and immunological factors, in order to predict who may benefit most from PRGF therapy.

This study demonstrates sex-based differences in PRGF molecular composition. In particular, the lower level of IL-1Ra and the higher level of P-selectin observed in males compared to females suggest in males a reduced anti-inflammatory buffering capacity [47] and greater platelet activation potential (pro-coagulant) [48]. Furthermore, PRGF reveals sex-dependent differences in EV surface marker profiles, particularly for CD2 and CD29, which were most prevalent in the male sample. CD2, which is typically found on T cells and natural killer cells, is involved in immune cell signaling, suggesting a potential immune-modulatory role for PRGF-derived EVs [49]. CD29 (integrin β1), on the other hand, has been associated with enhanced chondrogenic potential in mesenchymal progenitor cells. It was demonstrated that CD29 is expressed on migratory chondrogenic progenitor cells within human osteoarthritic cartilage, indicating a reparative phenotype [50]. Although the sex parameter is not statistically significant in terms of treatment responsiveness, the proteins and the EV profile indicate that males are more likely to respond successfully to treatment than females. These findings reinforce the concept that sex is a biologically significant variable in regenerative medicine, highlighting the need for sex-specific approaches in research and therapeutic applications.

The analysis conducted enabled the development of logistic regression models only at 2-month-follow-up for lack of sufficient data at the other follow-ups.

The model identified covariates that could play an important role in determining the observed outcome in our study. Among the variables included in the analysis, six were found to be suitable for inclusion in a multivariate logistic regression model. Specifically, BMI, biceps fold, fat mass percentage, red blood cells, platelets, and neutrophil counts approached statistical significance, further supporting the potential influence of baseline platelet and blood cell counts, as well as body composition, on the treatment outcome’s responsiveness rate [19]. However, it should be noted that statistical significance does not automatically imply a causal relationship but rather a consistent association with the outcome of our study.

Moreover, the model demonstrated a very high AUC value, indicating excellent discriminatory ability between positive and negative cases. This result is particularly relevant, as it suggests that the models not only correctly identify significant variables, but they also can be effective as predictive tools in future practical applications.

Despite the promising results, the analysis has some limitations. The limited number of patients included in this study prevents the identification of specific metabolic or nutritional factors that could differentiate between responsive and non-responsive patients to PRP (or PRGF) treatment for knee OA. To address the potential risk of overfitting in the model derived from our dataset, the Elastic Net regularization technique integrated with a nested LOOCV cross-validation method was applied. This approach resulted in the development of a more robust and reliable model, which decreased its predictive power, while maintaining good AUC value. Nevertheless, the generalizability of these models remains dependent on the representativeness of the sample, which should be expanded and diversified in subsequent studies.

Consequently, future studies involving larger patient cohorts would be necessary to address this limitation and eventually further validation is required to ensure the effectiveness of the generated models in different contexts.

Several limitations were identified in our study. A major limitation is the increasing loss at follow-up, which progressively reduced the number of patients. This decline may have introduced bias and limited the generalizability of our findings. Given the lack of information to support valid imputation, a complete-case approach has been exploited, analyzing only the available data at each time point. This strategy may have underestimated or overestimated some of the results. Secondly, the relatively small cohort size, particularly after stratification into subgroups, limits statistical power and may reduce the reliability of the analyses. The post hoc power analysis confirmed that the study was underpowered to detect small to moderate effects of the subgroup analyses. Nonetheless, given the exploratory and pilot nature of work, the report of these findings aimed to provide initial insights and generate hypotheses for future and larger studies.

Finally, the evaluation of the responsiveness in this study was exclusively on non-imaging, patient-reported clinical outcomes which may not fully capture structural changes. Future studies should objectively evaluate intra-articular changes in patients with knee OA after treatment with PRGF.

## 5. Conclusions

The identification of metabolic and immunological factors influencing PRGF composition and treatment response highlights the importance of personalized approaches in the management of OA. The development of logistic regression models at two-month follow-up identified body composition and blood cell count at baseline as key covariates as potential predictors of treatment response. Future studies on large patient cohorts focusing on optimizing PRGF protocols, understanding the role of EVs, and exploring patient-specific metabolic and immunological factors will be essential to maximize the therapeutic potential of PRGF in OA.

## Figures and Tables

**Figure 1 nutrients-17-03134-f001:**
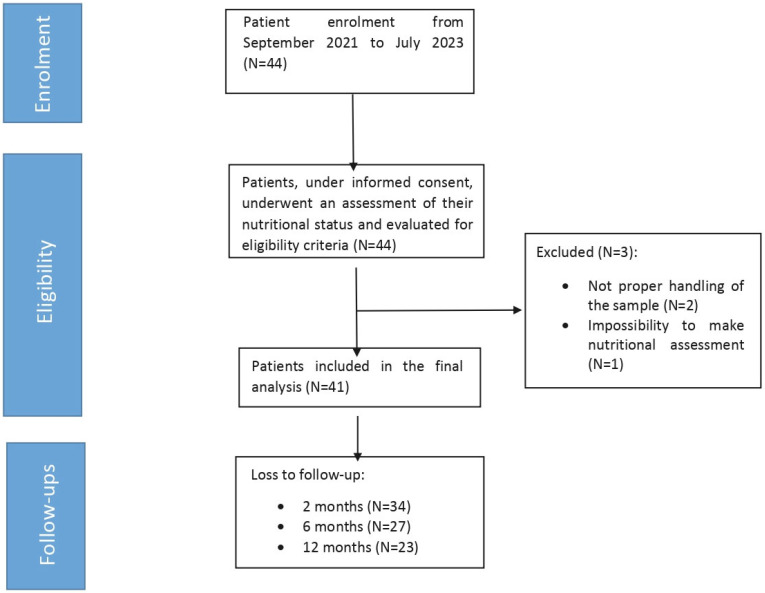
STROBE-like participant flow diagram reporting the enrolment, eligibility assessment, exclusions, and loss to follow-up reduction.

**Figure 2 nutrients-17-03134-f002:**
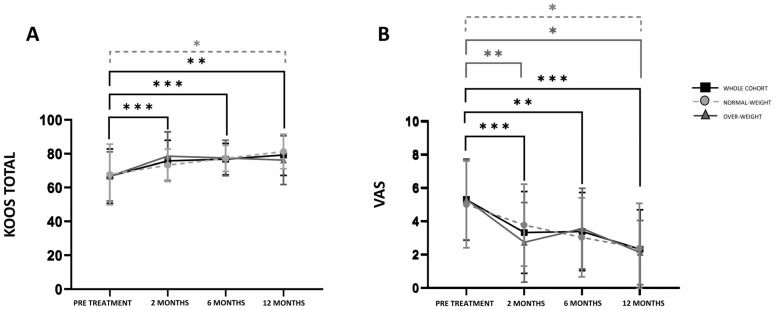
KOOS total (**A**) and VAS scores (**B**) of the whole cohort, normal weight, and overweight patients at pre-treatment, 2, 6, and 12 months of follow-up. * = *p*-value < 0.05, ** = *p*-value < 0.01, *** = *p*-value < 0.001.

**Figure 3 nutrients-17-03134-f003:**
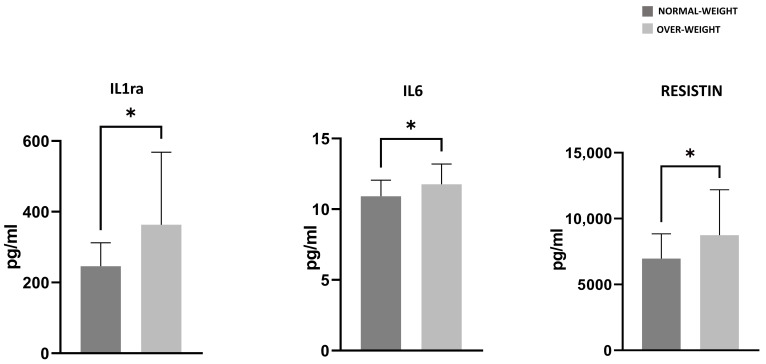
IL1ra, IL6, and resistin levels measured in PRGF of normal weight (dark gray bars) and overweight (light gray bars) patients expressed in pg/mL. Adjusted *p*-values (Benjamini–Hochberg FDR correction) are reported. * = *p*-value < 0.05.

**Figure 4 nutrients-17-03134-f004:**
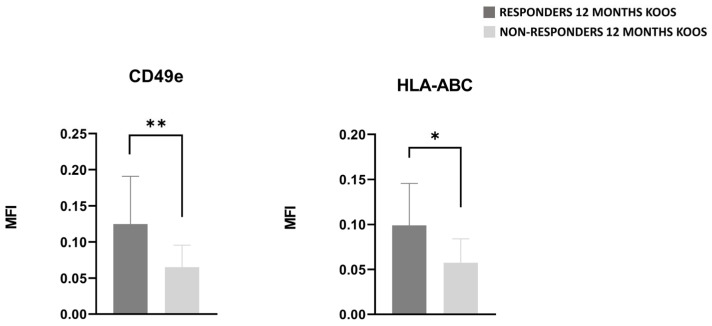
CD49e and HLA-ABC detected in PRGF’ EVs of responder (dark gray bars) and non-responder (light gray bars) patients 12 months of follow-up. Results are expressed in MFI. Adjusted *p*-values (Benjamini–Hochberg FDR correction) are reported. * = *p*-value < 0.05, ** = *p*-value < 0.01.

**Figure 5 nutrients-17-03134-f005:**
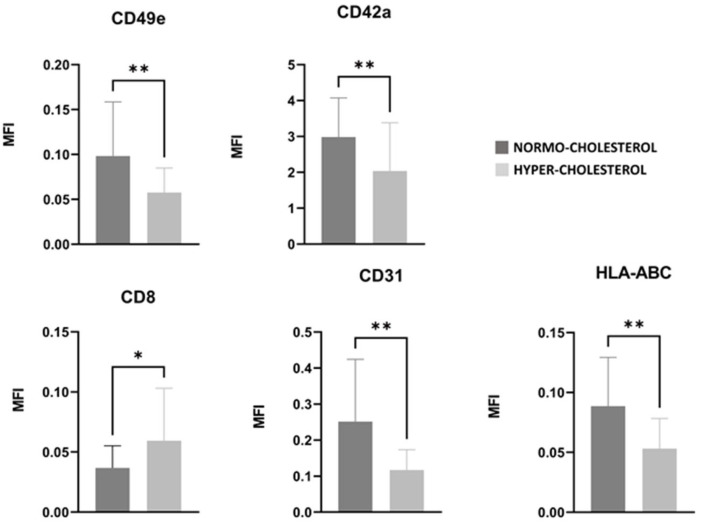
CD8, CD49e, HLA-ABC, CD42a, and CD31 detected in PRGF’ EVs of normocholesterolemic (dark gray bars) and hypercholesterolemic (light gray bars) patients. Results are expressed in MFI. Adjusted *p*-values (Benjamini–Hochberg FDR correction) are reported. * = *p*-value < 0.05, ** = *p*-value < 0.01.

**Figure 6 nutrients-17-03134-f006:**
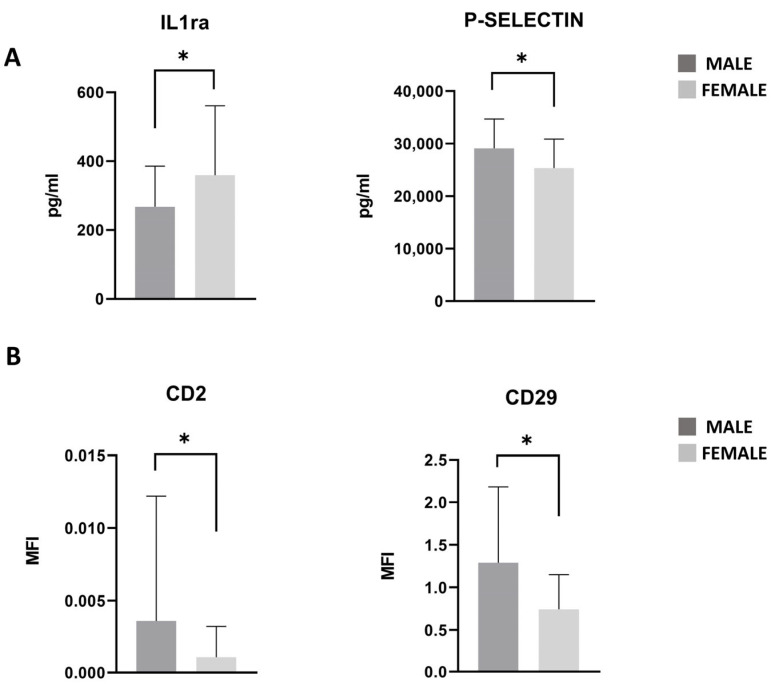
IL1ra and P-Selectin levels in PRGF of male (dark gray bars) and female patients (light gray bars). Results are expressed in pg/mL (**A**). CD2 and CD29 detected in PRGF’ EVs of male (dark gray bars) and female patients (light gray bars) (**B**). Results are expressed in MFI. Post hoc *t*-tests adjusted for Benjamini–Hochberg FDR correction are reported. * = *p*-value < 0.05.

**Figure 7 nutrients-17-03134-f007:**
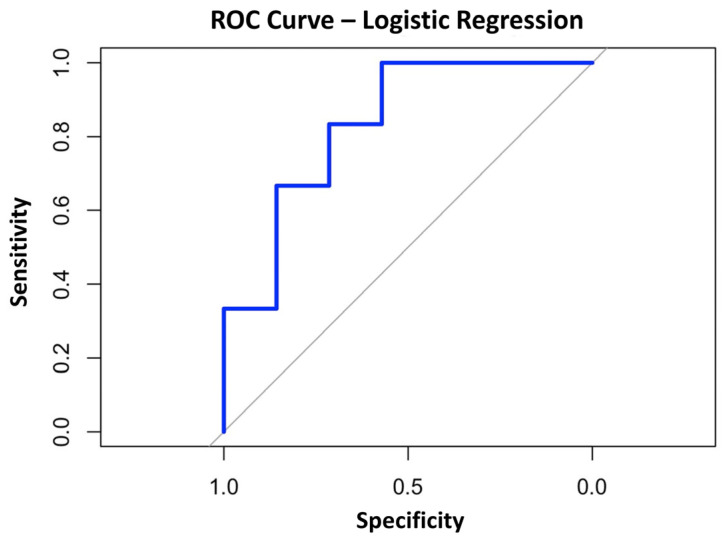
Receiver operating characteristic (ROC) curve resulting from Logistic Regression modeling. Red dashed line represents the ROC curve for a random guess, whereas the area under the curve (AUC) corresponds to 0.833 (95% CI: 0.461–0.949).

**Figure 8 nutrients-17-03134-f008:**
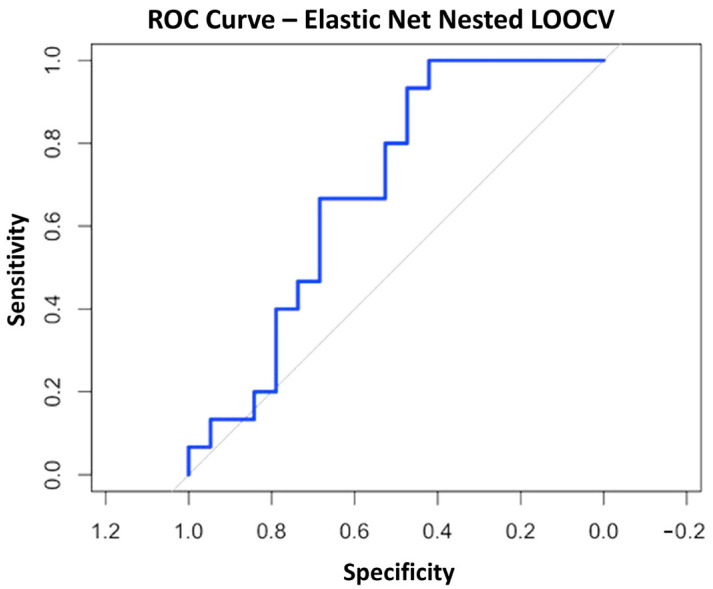
Receiver operating characteristic (ROC) curve resulting from the adjustment of logistic regression model through an Elastic Net regularization approach and after implementation of nested cross-validation as external validation set. The AUC value decreased to 0.691 (95% CI: 0.51–0.87). This indicates a reduction in predicting accuracy but results in a more reliable model.

**Table 1 nutrients-17-03134-t001:** Anthropometric characteristics of participants.

ID	Age (Years)	Sex	Weight (kg)	Height (m)	BMI (kg/m^2^)	Waist Circumference (cm)	Arm Circumference (cm)	Biceps Fold (mm)	Triceps Fold (mm)	Subcrestal Fold (mm)	Subscapular Fold (mm)	FFM %	FM%	MD
ID 1	47	M	68.5	1.69	24.1	87.5	29	4	4.8	9.8	15	80.9	19.1	12
ID 2	32	M	77	1.76	25.0	99.5	30.5	7.2	9.2	22.4	18.6	77	23	10
ID 3	53	F	51.5	1.66	18.7	67	25.5	3	7	5	7.5	77.1	22.9	11
ID 4	54	F	51	1.63	19.2	72	24.5	4	13.8	11.2	11.4	69.6	30.4	11
ID 5	47	M	107	1.81	32.7	108	36	8.2	17.8	36	29.2	67	33	11
ID 6	68	F	85	1.62	32.4	106	33	8.2	13	20	22.4	63.6	36.4	8
ID 7	50	M	76	1.77	24.3	91	34.5	8	12	20.4	17.8	71.3	28.7	11
ID 8	48	M	79	1.78	24.9	96	32.5	6.6	12.8	25.8	18.8	72	28	10
ID 9	44	F	68	1.60	26.7	86	32	5.2	25	27.2	23	63.1	26.9	9
ID 10	29	M	87	1.77	27.8	96.5	33.5	7	16	27	15	77.9	22.1	10
ID 11	41	M	78.5	1.78	24.8	92	32	5.4	14.8	14	12.4	76.4	23.6	11
ID 12	24	M	72.5	1.72	24.5	92	31	4.2	20	21.4	18.6	78.1	21.9	12
ID 13	57	F	60.5	1.68	21.6	77	26	6.6	16	20.4	14.6	64.9	35.1	11
ID 14	65	M	85	1.75	27.9	102	33	4	15.6	23.6	27.8	68.1	31.9	4
ID 15	60	M	72.5	1.68	25.6	96.5	31	4.2	5.2	15.4	14.6	77.3	22.7	9
ID 16	51	M	71	1.93	19.2	84	30	3.2	8.8	10.4	7.2	81.6	18.4	10
ID 17	35	M	80	1.88	22.6	82	30	3.4	6.8	33	10.6	84.6	15.4	13
ID 18	34	M	81	1.73	27.1	85.5	38.5	5	6.4	20.6	11.4	80	20	5
ID 19	65	M	94	1.78	29.8	103	33	6.4	14.8	21.8	15.6	71.2	28.8	12
ID 20	48	F	60	1.64	22.4	85.5	27.5	5.8	13.8	16.8	13.6	69.1	30.9	12
ID 21	51	F	50	1.54	19.4	71	25.5	6.2	14.2	20.2	11.6	66.2	33.8	12
ID 22	51	M	86.5	1.77	27.8	94.5	35.5	5.8	14.4	25.8	13	71.1	28.9	11
ID 23	64	M	72	1.68	25.5	91.5	31.5	3.8	16.8	19.4	16.4	71.8	28.2	11
ID 24	65	F	76	1.67	27.2	96	28.5	6.4	24	21	20.8	61.8	38.2	11
ID 25	57	M	81	1.80	25.1	93	34	3.2	9	18.4	12.8	75.8	24.2	11
ID 26	54	F	68	1.62	25.9	97	31	4.4	18.6	22.2	12.8	64.8	35.2	8
ID 27	44	M	76	1.80	23.5	84	32.5	3.8	5.8	10	10.4	82.4	17.6	7
ID 28	59	F	62	1.60	24.1	82.5	32.5	14	31	20	19.6	59.7	40.3	14
ID 29	45	M	87	1.79	27	96.5	36	7	20.3	18.6	15	72.7	27.3	14
ID 30	41	M	80	1.85	24	92	33.5	5	14	27	12.8	73.2	26.8	12
ID 31	61	F	71.5	1.57	28.8	106.5	32.5	16	24.8	23	18	60.2	39.8	11
ID 32	48	F	57	1.53	24.3	80.5	29.5	14.4	26.2	25	13.2	66.4	36.6	/
ID 33	52	M	70	1.82	21.13	79	28.5	4.2	11.4	8	10.6	79.5	20.5	12
ID 34	44	M	65	1.68	23.17	88.5	32	5.4	13.6	23	11.4	72.6	27.4	14
ID 35	48	M	93	1.78	29.4	102.5	39.5	2	23.4	19.4	14.4	73.1	26.9	11
ID 36	52	F	72	1.60	29.1	89	31.5	21.4	26.2	21.4	24.2	58.4	41.6	10
ID 37	45	M	71.6	1.18	23	91.5	32	8.8	18.6	18	8.4	74.4	25.6	12
ID 38	56	F	80	1.63	30.3	103	33	14.8	34	31	25	56.8	43.2	13
ID 39	59	M	97	1.75	31.5	106	40	9.2	15	19.2	16	71	29	4
ID 40	52	F	60.45	1.63	22.7	77	28.5	6	16	9	12	68.7	31.3	9
ID 41	62	F	50	1.65	18.4	72	24	3.6	16	11	9.2	69.7	30.3	13

BMI: Body mass index, FFM%: percentage of free fatty mass, FM: fatty mass, MD: Mediterranean diet index.

**Table 2 nutrients-17-03134-t002:** Variables included in the logistic regression model and their parameter estimates.

Coefficients	Estimate	STD Error	z Value	*p* Value
Intercept	−3.8435	6.4810	−0.5930	0.5532
BMI	0.2308	0.1703	1.3560	0.1753
Biceps fold (mm)	−0.5069	0.2677	−1.8940	0.0583 #
FM%	0.2086	0.1181	1.7660	0.0774 #
Red blood cells count	−2.5749	1.3391	−1.9230	0.0545 #
Platelets count	0.0149	0.0094	1.5820	0.1136
Neutrophils cells count	1.0055	0.5216	1.9280	0.0539 #

BMI: Body mass index, FM%: percentage of fatty mass. # corresponds to tendencies close to *p*-value significance.

## Data Availability

The protocol was prospectively registered with Clinicaltrials.gov (Identifier number NCT05546541). The datasets presented in this study can be found in online repositories with link: https://osf.io/hrcym/?view_only=9ba99de8cad14d8ca72d5f11c455a6b2 (accessed on 28 September 2025).

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
