# Peer review of "Impact of Nutritional Status on Clinical Outcomes of Patients Undergoing PRGF Treatment for Knee Osteoarthritis—A Prospective Observational Study"

_nutrients, 2025, doi:10.3390/nu17193134_

Round 1

Reviewer 1 Report

Comments and Suggestions for Authors

Dear Authors,

I appreciate the prospective nature of your case series and the clinically relevant aim of linking metabolic–nutritional status with PRGF characteristics and patient-reported outcomes in knee osteoarthritis. The use of validated PROMs (KOOS, VAS), trial registration and ethical approval, and your in-depth profiling of cytokines and extracellular vesicles (EVs) are clear strengths. Before this manuscript can be considered further, however, several aspects of reporting and methodology need substantial clarification and tightening. First, I encourage you to present a transparent participant flow (STROBE-style), with the exact numbers analysed at 2, 6, and 12 months, and an explicit strategy for handling missing data; loss to follow-up should be quantified and its impact assessed. Second, the manuscript reports numerous comparisons across EV markers, cytokines, and subgroups at α=0.05, but no explicit multiplicity control; please apply an FDR procedure (e.g., Benjamini–Hochberg) and update the text, tables, and figures accordingly. Third, regarding the predictive modelling, the initially high AUC, accompanied by wide confidence intervals, suggests overfitting; after regularization, the performance is moderate (AUC ≈0.70). I ask you to prioritize the regularized model, report uncertainty comprehensively, and, if feasible, implement nested cross-validation and/or provide external validation. Fourth, interpretation of subgroup findings (e.g., BMI, sex, hypercholesterolaemia) should be tempered and adjusted for plausible confounders such as concomitant medications (NSAIDs/SYSADOA), physiotherapy, physical activity, and comorbid metabolic conditions. Fifth, please expand the clinical results by presenting absolute KOOS changes (means, SD/95% CI) for the total score and subscales, and state clearly which endpoint was pre-specified as primary in the registered protocol. Sixth, in Methods 2.3 (PRGF Collection), you define F1 (PPP) and F2 (PRP); CaCl₂-activated F2 is injected intra-articularly, whereas F1 is activated “for research purposes” and subsequently characterized. This means the biologically characterized material is not identical to the injected product. Please justify why F1 can serve as a valid proxy for F2, provide comparative data (F1 vs F2), and/or replicate key cytokine/EV profiling on F2. Seventh, please ensure consistent terminology throughout (e.g., “normal-weight/overweight,” “follow-up”) and specify units and post-hoc procedures in figure legends to facilitate reproducibility. Eighth, it would be helpful to elevate selected laboratory variables from Appendix/Supplementary Table A1 into the main text, ideally stratified by responder status, to support the clinical narrative. Finally, while the preliminary signals you report are intriguing and consistent with the hypothesis of immunometabolic modulation of PRGF effects, the absence of a control group and the small sample size limit causal inference; framing the study explicitly as exploratory/pilot and moderating the strength of conclusions will improve alignment between data and claims. I believe that addressing the above issues—especially the comparability of the analysed and injected fractions (F1 vs F2), multiplicity control, transparent handling of missing data, and more conservative modelling—will substantially strengthen your work's scientific soundness and interpretability. Please provide a detailed, point-by-point response explaining how each concern has been addressed and where revisions are reflected in the manuscript.

Best regards,

The reviewer.

Author Response

The authors would like to thank the reviewers for their valuable comments, which have helped to enhance the quality of the manuscript. Each response has been addressed point by point. The changes can be seen in review mode.

Reviewer 1

Dear Authors,

I appreciate the prospective nature of your case series and the clinically relevant aim of linking metabolic–nutritional status with PRGF characteristics and patient-reported outcomes in knee osteoarthritis. The use of validated PROMs (KOOS, VAS), trial registration and ethical approval, and your in-depth profiling of cytokines and extracellular vesicles (EVs) are clear strengths.

Before this manuscript can be considered further, however, several aspects of reporting and methodology need substantial clarification and tightening.

First, I encourage you to present a transparent participant flow (STROBE-style), with the exact numbers analysed at 2, 6, and 12 months, and an explicit strategy for handling missing data; loss to follow-up should be quantified and its impact assessed.

R: We thank the Reviewer for this indication. In response, we added a STROBE-style participant flow diagram (Figure 1, from now the number of figures shifted), reporting the number of participants analysed at 2, 6 and 12 FU in the first paragraph of results section. We also described explicitly the handling of loss to follow up on the materials and methods section (lines 107-109), and the section addressing this topic in the limitation part was expanded (lines 575-577).

Second, the manuscript reports numerous comparisons across EV markers, cytokines, and subgroups at α=0.05, but no explicit multiplicity control; please apply an FDR procedure (e.g., Benjamini–Hochberg) and update the text, tables, and figures accordingly.

R: The authors appreciate the reviewer’s observation. The Benjamini-Hocberg FDR procedure has been applied to the multiple t-tests. The materials and methods has been supplemented with this statistical method (lines 217-219) and text, tables and figures have been modified accordingly.

Third, regarding the predictive modelling, the initially high AUC, accompanied by wide confidence intervals, suggests overfitting; after regularization, the performance is moderate (AUC ≈0.70). I ask you to prioritize the regularized model, report uncertainty comprehensively, and, if feasible, implement nested cross-validation and/or provide external validation.

R: We thank the Reviewer for the accurate observation. As suggested, we implemented the regularized Elastic Net model through a nested cross-validation, using the inner layer set for tuning of the hyperparameters and the outer layer as the external validation set. Materials and methods section has been modified accordingly at lines 236-241 and reference 21 has been added to provide the methodology used. Results section has been modified accordingly at lines 433-439.

Fourth, interpretation of subgroup findings (e.g., BMI, sex, hypercholesterolaemia) should be tempered and adjusted for plausible confounders such as concomitant medications (NSAIDs/SYSADOA), physiotherapy, physical activity, and comorbid metabolic conditions.

R: We thank the reviewer for raising this point. While we have information on medication use and physiotherapy treatments, the number of patients in each subgroup is too low for stratification to be feasible. Regarding metabolic alterations, all these were already considered in our analysis, as described in the text.

Fifth, please expand the clinical results by presenting absolute KOOS changes (means, SD/95% CI) for the total score and subscales, and state clearly which endpoint was pre-specified as primary in the registered protocol.

R: The authors thank the reviewer for the request. We have now included the primary outcome and table with KOOS subscales for each patient at each time point investigated in the supplementary material (lines 297-298).

Sixth, in Methods 2.3 (PRGF Collection), you define F1 (PPP) and F2 (PRP); CaCl₂-activated F2 is injected intra-articularly, whereas F1 is activated “for research purposes” and subsequently characterized. This means the biologically characterized material is not identical to the injected product. Please justify why F1 can serve as a valid proxy for F2, provide comparative data (F1 vs F2), and/or replicate key cytokine/EV profiling on F2.

R: The authors thank the reviewer for highlighting an important point in the study that may require further clarification. In this study, the F1 fraction of each patient was analysed in vitro, particularly the lower portion (i.e. the portion closest to the underlying F2). This is because, for ethical reasons, it is not possible to remove significant portions of the F2 fraction from the patient as it is required for infiltration into the treatment site. This is a very common procedure in this type of study. Previous studies show that only certain factors may be more diluted in the F1 fraction [doi: 10.1080/09537104.2017.1319046]. Furthermore, any differences remain proportional in all samples analysed, and the significance between the different subgroups is not affected.

Seventh, please ensure consistent terminology throughout (e.g., “normal-weight/overweight,” “follow-up”) and specify units and post-hoc procedures in figure legends to facilitate reproducibility.

R: The authors thank the reviewer and have standardised the text in the results section and added the post hoc test to the caption of Figure 6, as requested.

Eighth, it would be helpful to elevate selected laboratory variables from Appendix/Supplementary Table A1 into the main text, ideally stratified by responder status, to support the clinical narrative.

R: In an earlier version of the manuscript, we included the blood data results from the table within the main text; however, this significantly reduced readability and disrupted the flow. Additionally, describing specific blood values in the text appeared forced, particularly given the challenge of clearly defining selection criteria. Therefore, we chose to present this information exclusively in table format to maintain clarity and fluidity.

Finally, while the preliminary signals you report are intriguing and consistent with the hypothesis of immunometabolic modulation of PRGF effects, the absence of a control group and the small sample size limit causal inference; framing the study explicitly as exploratory/pilot and moderating the strength of conclusions will improve alignment between data and claims. I believe that addressing the above issues—especially the comparability of the analysed and injected fractions (F1 vs F2), multiplicity control, transparent handling of missing data, and more conservative modelling—will substantially strengthen your work's scientific soundness and interpretability. Please provide a detailed, point-by-point response explaining how each concern has been addressed and where revisions are reflected in the manuscript.

Reviewer 2 Report

Comments and Suggestions for Authors

< !--StartFragment -->

Major Comments:

The study is a prospective observational case series lacking a control group. This constrains causal inference concerning the efficacy of PRGF therapy relative to placebo or normal treatment. Kindly provide a more robust justification for the absence of a control arm and examine the consequences for interpretation.

The cohort size (n=41) is somewhat limited, especially when patients are categorized into subgroups (normal vs overweight, hypercholesterolemic vs normocholesterolemic, male vs female). The diminished statistical power raises apprehensions regarding the reliability of subgroup analysis. Kindly incorporate a power analysis and elucidate how this constraint influences the conclusions drawn.

Only patient-reported outcomes, specifically the KOOS and VAS, were evaluated. No imaging, biochemical cartilage indicators, or functional performance assessments were incorporated. This undermines the assumption that PRGF influences disease progression. Authors must to openly note this restriction and, if feasible, provide objective outcome measures for subsequent investigations.

Logistic regression and Elastic Net methods were utilized on a limited dataset, heightening the danger of overfitting. The reported elevated AUCs may lack generalizability. Additional information is required regarding the validation of the models, particularly the constraints of leave-one-out cross-validation in small cohorts.

The biological interpretation of differential EV markers (e.g., CD86, CD49e, HLA-ABC) is still conjectural, despite their intrigue. The discourse must distinctly differentiate statistically significant results from trends and refrain from overinterpretation absent mechanical validation.

The dietary anamnesis and evaluation of the Mediterranean diet index are succinctly outlined; nonetheless, the rigor of their application or validation within this cohort remains ambiguous. Additional information regarding reproducibility, potential recall bias, and the standardization of dietary advising is required.

The study asserts originality in correlating dietary and metabolic parameters with the efficacy of PRGF. Nonetheless, several aspects of the discourse reaffirm previously established correlations among fat, inflammation, and osteoarthritis. Kindly highlight what is distinctly illustrated here, in addition to validating established risk factors.

Minor Comments

The abstract is dense; simplifying sentences and better highlighting the main novel findings would improve clarity.

The manuscript alternates between “PRGF” and “PRP.” Ensure consistent terminology throughout to avoid confusion.

Figures (e.g., Figures 2–5) should include sample sizes (n) in legends, and axis labels should be clearer (units, abbreviations).

Several grammatical errors and awkward phrasings (e.g., “demonstrated a significative decrease,” “was obtain”) should be corrected for fluency. A professional language edit is recommended.

Some references are dated or general; the discussion would benefit from inclusion of the most recent systematic reviews/meta-analyses on PRP/PRGF in OA.

The clinical trial identifier is provided (NCT05546541). It would strengthen the manuscript to report whether the trial was prospectively registered and whether all pre-specified outcomes were presented.

< !--EndFragment -->

Author Response

The authors would like to thank the reviewers for their valuable comments, which have helped to enhance the quality of the manuscript. Each response has been addressed point by point. The changes can be seen in review mode.

Reviewer 2

Major Comments:

The study is a prospective observational case series lacking a control group. This constrains causal inference concerning the efficacy of PRGF therapy relative to placebo or normal treatment. Kindly provide a more robust justification for the absence of a control arm and examine the consequences for interpretation.

R: Since PRGF treatment is currently widely used for patients with knee osteoarthritis and PRGF is a blood-derived product similar to PRP, the authors hypothesise that patients' metabolic status may significantly influence the quality of the product and clinical outcomes. This study does not aim to assess the efficacy of PRGF, which has been extensively investigated in comparison with control groups, but rather to evaluate how the metabolic and nutritional status of patients with knee osteoarthritis undergoing PRGF treatment affects product quality and, more importantly, clinical outcome. Therefore, according to the study design, including a control group treated with hyaluronic acid, corticosteroids, or other therapies is deemed neither appropriate nor relevant.

The cohort size (n=41) is somewhat limited, especially when patients are categorized into subgroups (normal vs overweight, hypercholesterolemic vs normocholesterolemic, male vs female). The diminished statistical power raises apprehensions regarding the reliability of subgroup analysis. Kindly incorporate a power analysis and elucidate how this constraint influences the conclusions drawn.

R: The authors see the reviewer’s observation. To address this concern, a post hoc sensitivity power analysis for each subgroup has been performed. The results of this analysis are displayed in the results section. Moreover, the analysis has been added to the Materials and Methods section and how this results have influenced the conclusion is addressed in the discussion/limitation (lines 575-585).

Only patient-reported outcomes, specifically the KOOS and VAS, were evaluated. No imaging, biochemical cartilage indicators, or functional performance assessments were incorporated. This undermines the assumption that PRGF influences disease progression. Authors must to openly note this restriction and, if feasible, provide objective outcome measures for subsequent investigations.

R: Thank you to the reviewer for this clarification. As this was an observational, non-interventional study, it was not possible to ask patients to return to hospital for objective assessments at various follow-up. However, an official digital system (Qualtrics) was used to collect patient-reported outcome measures (PROMs) from each patient, including the VAS pain scale and the KOOS scale, which assesses pain, symptoms, activity, sports and quality of life. We have now added the KOOS subscales table to the article. Many studies to date rely on these scales to assess knee osteoarthritis in patients. Undoubtedly, the possibility of performing an MRI scan before treatment and at the final follow-up, when permitted by the ethics committee, adds significant value. This limitation is mentioned in the text.

Logistic regression and Elastic Net methods were utilized on a limited dataset, heightening the danger of overfitting. The reported elevated AUCs may lack generalizability. Additional information is required regarding the validation of the models, particularly the constraints of leave-one-out cross-validation in small cohorts.

R: As already answered to Reviewer’s 1 concerns, we added the implementation of a nested LOOCV. The inner layer set was used for tuning of the hyperparameters and the outer layer as the external validation set, allowing a more robust building and validation of the model. This methodological specification has been clarified in Materials and Methods section.

The biological interpretation of differential EV markers (e.g., CD86, CD49e, HLA-ABC) is still conjectural, despite their intrigue. The discourse must distinctly differentiate statistically significant results from trends and refrain from overinterpretation absent mechanical validation.

R: In accordance to the reviewer's comment, while our results reveal trends that provide valuable insights into the behavior of EV markers, our discussion focuses exclusively on data that reached statistical significance.

The dietary anamnesis and evaluation of the Mediterranean diet index are succinctly outlined; nonetheless, the rigor of their application or validation within this cohort remains ambiguous. Additional information regarding reproducibility, potential recall bias, and the standardization of dietary advising is required.

R: The authors clarify that, for the present study, a detailed dietary history was obtained, and validated tools such as the Mediterranean Diet Adherence Screener were employed to assess adherence to the Mediterranean diet. Additionally, the QMV method was used to calculate the average daily caloric intake, as previously described in the literature [G Tarrini, S Di Domizio, R Rossini, A Romano, F Cerrelli, G Marchesini Reggiani, et al. (2006). Quanto mangio veramente?. GIDM. GIORNALE ITALIANO DI DIABETOLOGIA E METABOLISMO, 26, 48-53]. Furthermore, all evaluations of patients’ nutritional status were conducted by the same nutritionist, as now explicitly indicated in the manuscript (lines 97,112).

The study asserts originality in correlating dietary and metabolic parameters with the efficacy of PRGF. Nonetheless, several aspects of the discourse reaffirm previously established correlations among fat, inflammation, and osteoarthritis. Kindly highlight what is distinctly illustrated here, in addition to validating established risk factors. 

R: This pioneering study investigates the impact of patients' nutritional and metabolic conditions on the quality of PRGF, associating this data with clinical outcomes for the first time. It therefore demonstrates that body composition and blood cell count at the time of treatment are potential predictors of response to treatment. While literature establishes a correlation between fat, inflammation and osteoarthritis, as the reviewer pointed out, there is no clinical evidence that an autologous blood derived treatment such as PRGF is effective in subjects with different nutritional and metabolic characteristics. This study lays the foundations for important future research aimed at optimising autologous treatments in regenerative medicine and reducing the proportion of patients who do not respond to treatment, with a view to achieving precision medicine.

Minor Comments

The abstract is dense; simplifying sentences and better highlighting the main novel findings would improve clarity.

R: We thank the reviewer for the request. The abstract has now been restructured.

The manuscript alternates between “PRGF” and “PRP.” Ensure consistent terminology throughout to avoid confusion.

R: We thank the reviewer for this clarification. We have now replaced PRP with PRGF where appropriate. Where the term PRP remains, we believe it is correct to leave it as it is, either because it derives from the trade name of the Endoret product or because it is associated with a reference.

Figures (e.g., Figures 2–5) should include sample sizes (n) in legends, and axis labels should be clearer (units, abbreviations).

R: All the figures are provided with legends and units; the only exception was for Figure 2. KOOS TOTAL and VAS have been moved from the label of the graph to the y axis to improve clarity. Sample sizes are always reported in the main text.

Several grammatical errors and awkward phrasings (e.g., “demonstrated a significative decrease,” “was obtain”) should be corrected for fluency. A professional language edit is recommended.

R: A grammar check of the full article has been executed.

Some references are dated or general; the discussion would benefit from inclusion of the most recent systematic reviews/meta-analyses on PRP/PRGF in OA.

R: As suggested, we have now added two recent references (DOI: 10.1016/j.arthro.2024.03.018; DOI: 10.1080/14712598.2025.2465833).

The clinical trial identifier is provided (NCT05546541). It would strengthen the manuscript to report whether the trial was prospectively registered and whether all pre-specified outcomes were presented.

R: The study protocol for this paper was prospectively registered on ClinicalTrials.gov. as reported in “Data Availability Statement” section. The portal has not yet been updated with the obtained results, which we will do as soon as possible. In addition to PRGF, which is the subject of this paper, the registered protocol includes investigations relating to microfragmented adipose tissue. However, the data will not be published at this time due to difficulties in recruiting patients, for reasons internal to the institute.

Round 2

Reviewer 1 Report

Comments and Suggestions for Authors

Dear Authors,

Thank you very much for your detailed answers and all the improvements you made.

Best regards,

The reviewer.

Reviewer 2 Report

Comments and Suggestions for Authors

The author improve well.